# Index Cancer Density Is a Stronger Predictor of Pelvic Lymph Node Invasion than Percentage of Biopsy-Positive Cores in EAU High-Risk Prostate Cancer: Clinical Impact in 254 Patients Treated and Staged with Robot-Assisted Radical Prostatectomy

**DOI:** 10.3390/cancers17203385

**Published:** 2025-10-21

**Authors:** Maria Angela Cerruto, Antonio Benito Porcaro, Alberto Bianchi, Alessandro Tafuri, Andrea Panunzio, Rosella Orlando, Francesca Montanaro, Alberto Baielli, Francesco Artoni, Andrea Franceschini, Lorenzo De Bon, Alessandro Veccia, Riccardo Rizzetto, Matteo Brunelli, Vincenzo De Marco, Filippo Migliorini, Salvatore Siracusano, Riccardo Giuseppe Bertolo, Alessandro Antonelli

**Affiliations:** 1Department of Urology, University of Verona, Azienda Ospedaliera Universitaria Integrata, 37129 Verona, Italy; antoniobenito.porcaro@aovr.veneto.it (A.B.P.); alberto.bianchi@aovr.veneto.it (A.B.); francesca.montanaro@univr.it (F.M.); alberto.baielli@univr.it (A.B.); francesco.artoni@univr.it (F.A.); andrea.franceschini@univr.it (A.F.); lorenzo.debon@univr.it (L.D.B.); alessandro.veccia@aovr.veneto.it (A.V.); riccardo.rizzetto@aovr.veneto.it (R.R.); vincenzo.demarco@aovr.veneto.it (V.D.M.); filippo.migliorini@aovr.veneto.it (F.M.); riccardogiuseppe.bertolo@univr.it (R.G.B.); alessandro.antonelli@univr.it (A.A.); 2Department of Urology, Vito Fazzi Hospital, 73110 Lecce, Italy; aletaf@hotmail.it (A.T.); panunzioandrea@virgilio.it (A.P.); rossella.orlando@asl.lecce.it (R.O.); 3Department of Pathology, University of Verona, Azienda Ospedaliera Universitaria Integrata, 37129 Verona, Italy; matteo.brunelli@univr.it; 4Department of Life, Health and Environmental Sciences, University of L’Aquila, 67100 L’Aquila, Italy; salvatore.siracusano@univaq.it

**Keywords:** prostate cancer, EAU high-risk prostate cancer, percentage of biopsy positive cores, percentage of biopsy positive cores relative to prostate volume (index cancer density), prostate volume, robot-assisted radical prostatectomy (RARP), pelvic lymph node invasion

## Abstract

This study investigates whether the density percentage of biopsy-positive cores (BPCs) relative to prostate volume, named Id-BPC, can predict pelvic lymph node invasion (PLNI) in patients with high-risk prostate cancer, undergoing robot-assisted radical prostatectomy. Id-BPC proved to be a stronger predictor of PLNI than BPC. This highlights its potential to improve clinical decision-making and inform the development of more accurate predictive models, such as nomograms and artificial intelligence-based tools. These could enhance outcomes and treatment planning for patients with high-risk prostate cancer.

## 1. Introduction

Clinical prostate cancer (PCa) being classified as high-risk is an issue of such epidemic magnitude that both the European Association of Urology (EAU) and National Comprehensive Cancer Network (NCCN) institutions are actively involved in dealing with this subject; accordingly, management may vary from watchful waiting (WW) when life expectancy is less than 10 years to active treatments including radical prostatectomy with pelvic lymph node dissection (PLND), more frequently performed through the robotic approach, to radical radiation combined with long-term androgen deprivation; likewise, surgery should be considered within a set of potential multi-modal treatments [1,2]. However, EAU high-risk PCa is not homogeneous for including categories with or without adverse surgical pathology features with the former associating with the risk of pelvic lymph node invasion (PLNI) when having adverse cancer biology; accordingly, clinical practice needs more effective predictors of adverse pathology features, which have prognostic implications; likewise, tumor molecular biology, although representing the way forward, is not yet the clinical standard due to it being expensive [3,4,5,6]. In high-risk PCa, percentage of biopsy-positive cores (BPCs) is the strongest predictor of the risk of PLNI, thus entering all nomograms dealing with the subject [1,2]. However, BPC has not yet been evaluated as a parameter related to prostate volume (PV); likewise, the ratio of BPC to PV defines an index called Id-BPC, expressing the density of biopsy tumor load relative to the volume of the gland; accordingly, we wanted to test the hypothesis that this index could show a stronger association than BPC with the risk of PLNI, which is also the worst prognostic factor of this category.

## 2. Materials and Methods

### 2.1. EAU High-Risk PCa Treated with Robotic-Assisted Radical Prostatectomy (RARP): Patient Population and Evaluated Parameters

Before 25 September 2024, we planned a retrospective observational study design, which was granted an exemption status from formal approval by our Internal Review Board (IRB), in order to analyze a cohort of 254 EAU high-risk patients treated with RARP from January 2013 to December 2021, without any prior treatment, including androgen deprivation. All patients, thus, were classified as belonging to the high-risk prognostic group and were staged by a CT scan including chest/abdomen/pelvis and total body scan of bones or by PET-PSMA, as recommended by EAU guidelines; moreover, all selected patients did not undergo any prior treatment. Prostate volume was measured by several operators through transrectal ultrasound in each patient who had at least 12 random systemic biopsies. The ultrasonograph computes the total volume by measuring the length (L), height (H) and width (W) of the gland and multiplying the product by a coefficient of π/6 (0.52), also known as the prolate ellipsoid formula [7]. The same formula can also be used in Magnetic Resonance Imaging (MRI) modalities for the estimation of prostate volume [7]. Moreover, subjects were evaluated for age (years), body mass index (BMI; kg/m^2^), physical status, prostate-specific antigen, PSA (ng/mL), PV, BPC (%), tumor stage (cT) and grade; likewise, surgical specimens were assessed for grade and stage of tumors as well as for dissected lymph nodes, which were counted and checked for cancer invasion [1,2]. Accordingly, each subject was assessed by the American Society of Anesthesiologists (ASA) score system before undergoing RARP that was performed by 5 experienced surgeons who also staged pelvic lymph nodes [1,2].

### 2.2. Model Description and Statistical Methods

We hypothesized a model in which tumor load densities at clinical presentation (Id-BPC) could associate with a greater risk of PLNI than only BPC; accordingly, the higher the values of Id-BPC, the more likely the independent occurrence of PLNI by concomitant adverse features including cancer grade and stage. Moreover, once our assumption was verified by analysis, a cutoff point was identified based on the median values of the overall population. Continuous and categorical variables were investigated for medians with interquartile ranges (IQRs) as well as for frequencies with percentages (%), respectively. The investigated population was stratified according to PLNI, which was assessed for predictors by the logistic regression model (univariate and multivariate analysis). The software used to run the analysis was IBM-SPSS version 26. All tests were two-sided with *p* < 0.05 considered to indicate statistical significance.

## 3. Results

### 3.1. Demography of EAU High-Risk PCa and Associations with the Risk of PLNI

The demography of the EAU high-risk population treated and staged with RARP is reported in Table 1, which also shows distributions of both BPC and Id-BPC with the former and the latter having a median (IQR) of 42.0 (27–60%) and 1.0 (0.5–1.7%/mL), respectively.

Overall, PLNI was detected in 23.2% of cases. Accordingly, patients with PLNI were more likely to present with higher percentages of adverse tumor grades (ISUP 4–5), PSA levels > 20 ng/mL and BPC; likewise, they were also more likely to have higher Id-BPC densities showing a stronger association (OR = 1.444; 95% CI: 1.055–1.958; *p* = 0.018) than BPC (OR = 1.026; 95% CI: 1.014–1.038; *p* < 0.0001); moreover, the cT and cN stages did not show any significant association at all. In the surgical specimen, PLNI was also more likely to occur for aggressive cancers presenting with higher grades (ISUP 4–5) and stages (SVI); likewise, the number of counted lymph nodes did not impact such risk.

### 3.2. The Stronger Impact of Id-BPC than BPC on the Risk of PLNI in EAU High-Risk PCa

Multivariate analysis of clinical factors predicting the risk of PLNI in surgically treated EAU high-risk PCa is illustrated in Table 2, which shows three different models according to the different selected variables: Model 1 (PSA > 20 (ng/mL; ISUP 4–5; BPC), Model 2 (PSA > 20 (ng/mL; ISUP 4–5; Id-BPC), and Model 3 (PSA > 20 (ng/mL; ISUP 4–5; Id-BPC ≥ 1). Because of potential multicollinearity between BPC and Id-BPC, we did not evaluate both factors simultaneously in multivariate models.

Accordingly, PLNI was independently predicted by ISUP 4–5, PSA > 20 (ng/mL), BPC as well as by Id-BPC; likewise, the association of Id-BPC was stronger (OR = 1.926; 95% CI: 1.246–2.977; *p* = 0.003) than BPC (OR = 1.028; 95% CI: 1.014–1.042; *p* < 0.0001).

Moreover, when cancer density was categorized by Id-BPC ≥ 1.0 versus < 1.0, the prediction was even stronger (OR = 3.535; 95% CI: 1.551–8.054; *p* = 0.003) for the close association with the adverse tumor stage including seminal vesicle invasion and PLNI, as detailed in Table 3.

## 4. Discussion

The natural history of surgically treated clinical PCa depends on recurrences that occur in about 25% of cases and impacts disease progression with 10-year mortality rates ranging from 1.2% to 13.7%; however, beyond standard preoperative and perioperative variables predicting disease progression, more clinical factors predicting the risk of PLNI are required in order to reduce heterogeneity, which is also an issue for each prognostic risk class and for the EAU high-risk class [1,2,8,9,10,11,12]. In our study, we have shown that PLNI is a critical event occurring in 23.2% of EAU high-risk patients who were treated and staged with robotic surgery; accordingly, 76.8% of patients had an organ-confined disease in the surgical specimen, thus showing a better prognosis when compared with the former group; likewise, index cancer density was a stronger predictor than standard BPC for PLNI, which is the worst event occurring in this subcategory of patients; moreover, when Id-BPC was dichotomized at the level of one, which represented the median level occurring to the investigated population, the risk of PLNI was more than threefold when compared with the control subgroup (Id-BPC < 1.0%/mL). All study assumptions were supported by standard evidence showing that the prostate cancer biology is closely related to the volume of the gland [13]; likewise, we assumed that at the same PSA levels with constant tumor grades and BPC rates, tumor biology may not be homogeneous to be more aggressive for higher Id-BPC values. As an example, patients presenting with the same PSA levels, same ISUP grade group, but 10% BPC may occult different tumors for volumes of 30 mL, 60 mL and 90 mL; accordingly, tumor load densities would be 0.33, 0.16 and 0.11%/mL, respectively; as a result, Id-BPC is higher in the first case when compared to the last two; accordingly, cancer biology is supposed to be more aggressive for the first case when compared to the other two because of the higher tumor load density; nevertheless, higher cancer density at biopsy may also be associated with more aggressive biology in the surgical specimen; moreover, the median value of Id-BPC was 1.0%/mL (IQR: 0.5–1.7) in the investigated population, thus suggesting this cutoff, which we considered stronger than others. Accordingly, our results represent new findings, which may have clinical implications in EAU high-risk disease. In high-risk PCa, PLNI is predicted by adverse surgical tumor features including adverse tumor grade (ISUP 4–5) and stage (SVI); accordingly, the combination of patterns increases such risk, which predicts unfavorable prognosis (biochemical recurrence or persistence, progression and disease specific survival), as well [1,2,14,15,16]. Likewise, surgery is largely performed in both localized and locally advanced high-risk disease with PLNI rates being between 1.9% and 37.4% with overall and cancer-specific survival at 10-years ranging from 58% to 84% and from 65% to 96.2%, respectively; moreover, radical prostatectomy has shown to be a safe and reasonable therapeutic approach; however, the variable definition of high-risk disease remains a controversial issue [1,2,17,18,19,20,21,22,23]. Nevertheless, both the number of removed and positively counted lymph nodes are pivotal for evaluating surgically treated high-risk PCa patients, who need to be anatomically staged through an extensive pattern field because the lymphatic drainage of the prostate extends beyond standard nodal templates; accordingly, a number of 20 pelvic nodes may address a guideline for a sufficient PLND holding that the cN1 stage is not always sustained by metastases, but by hyperplastic or regressive alterations, as well [24,25,26,27,28].

In our study, we have demonstrated that index density of BPC was a stronger predictor than BPC of the risk of PLNI in the EAU high-risk population; accordingly, as Id-BPC increased, patients were more likely to have pelvic lymph node metastases; equivalently, as the cancer density index decreased, subjects were less likely to have an adverse pathology stage in the surgical specimen; nevertheless, the clinical tumor and nodal stage did not associate with such risk; likewise, cases presenting with Id-BPC ≥ 1.0 (%/mL) had a three-and-half-fold-higher risk of PLNI when compared with controls (Id-BPC less than 1.0%/mL), independently, and also by other adverse clinical features including ISUP 4–5 and PSA > 20 ng/mL. Because of potential multicollinearity between BPC and Id-BPC, we did not evaluate both factors simultaneously in multivariate models; likewise, Id-BPC was always a stronger predictor than BPC in both univariate and multivariate models, as shown in Table 1 and Table 2, respectively. These results have clinical implications when assessing this prognostic category of patients; accordingly, beyond standard adverse clinical parameters, high-risk PCa patients can further be stratified according to index cancer density levels that could be categorized at a reference point of 1.0 (%/mL) representing the median level detected in the overall population of whom 76.8% did not show any evidence of PLNI; moreover, these results represent the subject for evaluating new nomograms and/or neural networks for artificial intelligence (AI) systems, as well; nevertheless, these findings need confirmatory studies. So far, our results have practical implications for managing EAU high-risk patients in daily practice as Id-BPC is a simple tool which may be applied when consulting patients; accordingly, we started from theoretical assumptions relating Id-BPC to the most adverse outcome after surgery; moreover, the study demonstrated that the cutoff of 1.0%/mL stratified two categories of subjects of whom the one with Id-BPC ≥ 1.0%/mL showed a 3.5-fold increase in PLNI, including 23% of the EAU high-risk population needing extended counseling with combined aggressive treatments, including extended pelvic lymph node dissection for accurate anatomical staging for surgery or androgen blockade associated extended radiation of both the prostate and the pelvis. EAU high-risk PCa is a pivotal prognostic class which occults a complex heterogeneous biological system needing stratification in order to deliver appropriate sequential combined treatments. Likewise, our study showed that 23% of the high-risk population harbored pelvic lymph node metastases which were predicted by standard clinical parameters including PSA > 20 ng/mL, ISUP > 3 and BPC; likewise, Id-BPC demonstrated stronger prediction than the latter in a multivariate model, as shown in Table 2; moreover, it allowed effective patient stratification when categorized at the median level.

Interestingly, these findings are critical for understanding the prostate cancer high-risk system; accordingly, high PSA levels related to the metastatic load at pelvic lymph nodes while undifferentiated cancer associated with adverse tumor stage, including extracapsular extension and seminal vesicle invasion which is closed to related to PLNI, according to the complex anatomy associated the lymphatic pathway originating from the prostate gland; likewise, PLNI was also predicted by both continuous and categorized Id-BPC which is a more impactful factor than BPC itself for being a bidimensional variable relating BPC to the volume of the prostate thus defining a more aggressive biology within the EAU high-risk system, as well. As a result, Id-BPC should be included in multivariate models which specifically evaluate the risk of PLNI in the prostate cancer high-risk population [29]; accordingly, it can be included in dedicated nomograms as well as in AI networks, thus impacting more on clinical decisions than those including only BPC as the machines are able to test and learn several patterns that may present effective solutions which need to be assessed by the multidisciplinary PCa team in order to plan appropriate and effective decisions when counseling patients presenting with the unfavorable prognostic risk class.

In PCa basic research studies, an inverse association between volume of the gland and cancer biology has been shown; accordingly, tumors were more likely to be detected in smaller prostates, which also showed more aggressive cancer when compared with larger glands; as a theory, impaired circulating testosterone levels may have a role in cancers developing and progressing in microenvironments of dysregulated prostate volumes [30,31,32,33,34,35]. Our study showed that cancer densities, which were related to the volume of the gland, impacted the risk of PLNI in the investigated high-risk population; accordingly, cancer densities occurring in small-sized prostates were more likely to associate with adverse tumor stage including seminal vesicle invasion and PLNI, thus confirming the findings of the referenced studies; accordingly, index tumor density levels may represent a new way for assessing cancer biology related to volume growth of the gland, as well. The study showed strengths because procedures were performed by trained surgeons and all surgical specimens were evaluated by our dedicated pathologist.

The limits of our study are related to the following: the retrospective nature, the several surgeons performing procedures and biopsy protocol consistency, the missed evaluation of multiparametric MRI (mpMRI), which was not available in all cases, the volume of the gland, which was not always measured in our institution. Prostate volumes were measured by several operators through the transrectal ultrasound and this might induce a bias; however, this method is the standard in all urological units for decades [7]; accordingly, the risk of relative error rates did not impact the measured outcomes of the study; nevertheless, prostate volume did not show any association with PLNI, as shown in Table 1. Concerning the biopsy protocol consistency, although BPC calculations could be affected by number and location of cores obtained based on biopsy protocol consistency, this issue was not evaluated because locations of obtained cores were not available in all cases; nevertheless, it was not the aim of the study. Regarding the lack of mpMRI, as largely known from the reported literature [1], nomograms including mpMRI findings are less powerful than those not including it in multicenter studies because of the variability related to the operator in defining, scoring and locating PI-RADS lesions; accordingly, this is a major issue which influences study results evaluating mpMRI parameters. MRI lacks sensitivity for direct detection of positive lymph nodes, and as a consequence, nomograms combining clinical data, systematic or MRI-targeted biopsy results and, for some of them, MRI findings have been used to estimate the risk of patients harboring positive lymph nodes [1]. Several underwent external validation [29,36,37,38,39]. However, they tend to show limited specificity, and a substantial proportion of patients may still be submitted to unnecessary lymphadenectomy, especially when the lymph node invasion prevalence is low [1].

## 5. Conclusions

In an EAU high-risk PCa population treated and staged with robotic surgery, index density of BPC was a stronger predictor of PLNI than BPC; accordingly, as Id-BPC increased, patients were more likely to have PLNI; equivalently, subjects presenting with an Id-BPC of less than one were 3.5 times less likely to have metastases to pelvic lymph nodes. Accordingly, this information has implications for clinical practice as well as for computing nomograms or patterns of artificial intelligence networks.

## Figures and Tables

**Table 1 cancers-17-03385-t001:** Demographics and risk of pelvic lymph node invasion (PLNI) in EAU high-risk prostate cancer (PCa) treated and staged with robot-assisted radical prostatectomy (RARP).

	Population	No PLNI	PLNI	Univariate Analysis OR (95% CI)	*p*-Value
**Numbers (%)**	254	195 (76.8)	59 (23.2)		
**Clinical factors**					
Age (years)	66 (61–71)	65 (60.2–70)	68 (62–72)	1.042 (0.993–1.094)	0.095
BMI (kg/m^2^)	25.8 (24.2–28.4)	25.6 (24–28.1)	26.3 (24.4–29.3)	1.043 (0.958–1.137)	0.33
ASA score 3	32 (12.6)	21 (10.8)	11 (18.6)	1.899 (0.856–4.211)	0.115
PV (mL)	40 (30–55)	40 (30–55)	43 (32–56)	1.006 (0.992–1.021)	0.374
PSA > 20 (ng/mL)	55 (21.7)	32 (16.4)	23 (39)	3.254 (1.706–6.209)	<0.0001
ISUP 4–5	156 (61.4)	112 (57.4)	44 (74.6)	2.174 (1.134–4.169)	0.019
cT 2/3	169 (66.5)	133 (68.2)	36 (61)	0.730 (0.399–1.335)	0.306
cN1	65 (25.6)	53 (27.2)	12 (20.3)	0.684 (0.337–1.389)	0.293
BPC (%)	42.8 (27–60)	39.4 (25–57.1)	53.8 (40–78.5)	1.026 (1.014–1.038)	<0.0001
Id-BPC (%/mL)	1.0 (0.5–1.7)	0.9 (0.5–1.5)	1.2 (0.8–2.0)	1.444 (1.055–1.958)	0.018
**Surgical pathology**					
PW (gr)	53 (45–66)	51.6 (44.1–65)	55 (50–70.8)	1.013 (0.999–1.027)	0.064
ISUP 4–5	154 (60.6)	102 (52.3)	52 (88.1)	6.773 (2.931–15.652)	<0.0001
ECE	37 (14.6)	30 (15.4)	7 (11.9)	2.279 (0.837–6.204)	0.107
SVI	77 (30.3)	38 (19.5)	39 (66.1)	10.026 (4.857–20.697)	<0.0001
R1	88 (34.6)	56 (28.7)	32 (54.2)	2.942 (1.616–5.354)	<0.0001
**Counted lymph nodes**	25 (10–32)	25 (20–31)	28 (20–34)	1.024 (0.999–1.056)	0.125

Legend: continuous variables are reported as medians (interquartile ranges) and categorical factors as frequencies (percentages); ISUP, International Society of Urologic Pathology tumor grade group formulation; Id-BPC, index density of percentage of biopsy-positive cores; see also Materials and Methods for abbreviations.

**Table 2 cancers-17-03385-t002:** Multivariate analysis of clinical models predicting the risk of pelvic lymph node invasion in 254 EAU high-risk PCa patients treated and staged with RARP.

	Model 1 (*)		Model 2 (*)		Model 3 (*)	
Statistics	OR (95% CI)	*p*-Value	OR (95% CI)	*p*-Value	OR (95% CI)	*p*-Value
PSA > 20 (ng/mL)	4.988 (1.706–6.209)	<0.0001	5.386 (2.369–12.246)	<0.0001	5.260 (2.313–11.962)	<0.0001
ISUP 4–5	3.651 (1.560–8.543)	0.003	3.906 (1.698–8.984)	0.001	3.642 (1.582–8.304)	0.002
BPC (%)	1.028 (1.014–1.042)	<0.0001				
Id-BPC (%/mL)			1.926 (1.246–2.977)	0.003		
Id-BPC ≥ 1 (%/mL)					3.535 (1.551–8.054)	0.003

Legend: BPC, percentage of biopsy-positive cores; Id-BPC, index density of BPC; ISUP, International Society of Urologic Pathology tumor grade formulation; OR, odds ratio; CI, confidence interval; (*), after adjusting for all other clinical factors.

**Table 3 cancers-17-03385-t003:** Risk of adverse pathology in 254 EAU high-risk PCa patients treated and staged with RARP categorized by Id-BPC ≥ 1 (%/mL).

Variables	OR (95% CI)	*p*-Value
PW(gr)	0.958 (0.942–0.974)	<0.0001
ISUP 4–5	1.333 (1.020–1.049)	0.265
ECE	2.585 (1.193–5.386)	0.016
SVI	3.432 (1.888–6.238)	<0.0001
R1	2.078 (1.216–3.551)	0.007
PLNI	2.301 (1.236–4.283)	0.009

Legend: Id-BPC, index density of biopsy-positive cores; OR, hazard ratio; CI, confidence interval; Id-BPC was categorized with reference to values less than one (univariate analysis); see also Materials and Methods.

## Data Availability

Data is contained within the article.

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
