# Peer review of "Index Cancer Density Is a Stronger Predictor of Pelvic Lymph Node Invasion than Percentage of Biopsy-Positive Cores in EAU High-Risk Prostate Cancer: Clinical Impact in 254 Patients Treated and Staged with Robot-Assisted Radical Prostatectomy"

_cancers, 2025, doi:10.3390/cancers17203385_

Round 1
Reviewer 1 Report
Comments and Suggestions for Authors
Recommendation: Major revision
While the study presents valuable findings, major revisions are recommended to address the retrospective design, lack of mpMRI validation, variability in prostate volume measurements, and potential confounding due to procedures performed by multiple surgeons; further clarification is needed regarding methodology consistency and patient selection, and the results should be validated in prospective or multicenter settings for stronger generalizability and clinical impact.
- Clarify the inclusion and exclusion criteria for patient selection, particularly regarding prior treatments and imaging availability.
- Detail the methodology of prostate volume measurement by transrectal ultrasound; discuss potential measurement errors and their impact.
- Explain the rationale behind choosing the cutoff value of 1.0 mL for Id-BPC categorization and whether alternative cutoffs were tested.
- Elaborate on biopsy protocol consistency, including the number and location of cores obtained, as this affects BPC calculations.
- Consider the impact of potential multicollinearity between BPC and Id-BPC in multivariate models.
- Tables lack explicit p-values for some comparisons; ensure all relevant statistics are reported.
- Discuss the clinical meaning of a 3.5-fold increase in PLNI risk with Id-BPC greater than 1.0 mL, including patient management implications.
- Verify that all cited references are up-to-date and cover the latest guidelines and studies relevant to high-risk prostate cancer.
- Include a more detailed institutional review board approval section, specifying approval date and reference number.
Author Response
Comment 1: While the study presents valuable findings, major revisions are recommended to address the retrospective design, lack of mpMRI validation, variability in prostate volume measurements, and potential confounding due to procedures performed by multiple surgeons; further clarification is needed regarding methodology consistency and patient selection, and the results should be validated in prospective or multicenter settings for stronger generalizability and clinical impact. 1. Clarify the inclusion and exclusion criteria for patient selection, particularly regarding prior treatments and imaging availability.Response 1:
We thank the reviewer for the queries which allow us to clarify and improve our manuscript whose limits have already outlined in the discussion section. Accordingly, we did not evaluate mpMRI finding for not being available in all case, as stated in the discussion section while reporting the limits of the study which did not have that aim; likewise, as largely known form the reported literature, nomograms including mpMRI findings are less powerful than those not including it in multicenter studies because of the variability related to the operator in defining, scoring and locating PI-RADS lesions; accordingly, this is major issue which biases study results evaluating mpMRI parameters, as well; anyhow, these considerations have been emphasized in the discussion section.
The aim of the study was to evaluate Id-BPC in EAU high risk PCa including patients who were staged according to guideline recommendations; accordingly, patients classified as belonging to the high-risk prognostic group and were staged by CT scan including chest/abdomen/pelvis and total body scan of bones or by PET-PSMA, as recommended by EAU guidelines; moreover, selected patients did not undergo any prior treatment, as well. These features have been integrated in the first paragraph of the materials and method section, accordingly.
Comment 2: Detail the methodology of prostate volume measurement by transrectal ultrasound; discuss potential measurement errors and their impact.
Response 2: Prostate volumes were measured by several operators through the trans-rectal ultrasound and this might induce a bias; however, this method is the standard in all urological units since decades (Lee, Urol Int 2007); accordingly, the risk of relative error rates did not impact on measured outcomes of the study; nevertheless, prostate volume did not show any association with PLNI, as shown in Table 1. These considerations have been added to the discussion section relative to the limits of the study.
Comment 3: Explain the rationale behind choosing the cutoff value of 1.0 mL for Id-BPC categorization and whether alternative cutoffs were tested.
Response 3:
Study assumptions were supported by standard evidences showing the prostate cancer biology is closely related to the volume of the gland; likewise, we assumed that at same PSA levels with constant tumor grades and BPC rates, tumor biology may not be homogenous for being more aggressive for higher Id-BPC values. As an example, patients presenting with same PSA levels, same ISUP grade group, but 10% BPC may occult different tumors for volumes of 30 mL, 60 mL and 90 mL; accordingly, tumor load densities will be 0.33, 0.16 and 0.11 %/mL, respectively; as a result, Id-BPC is higher in the first case when compared to the last two; accordingly, cancer biology is supposed to be more aggressive for the first case when compared to the other two because of the higher tumor load density; nevertheless, higher cancer density at biopsy may associate with more aggressive biology in the surgical specimen, as well; moreover, median value of Id-BPC was 1.0 %/mL (IQR: 0.5 – 1.7) in the investigated population thus suggesting as to test this cutoff, which we retained stronger than others, as well.
Thus, the rationale behind the chosen cutoff has been added to the second paragraph of the method section and all these features have been inserted into the discussion as well.
Comment 4: Elaborate on biopsy protocol consistency, including the number and location of cores obtained, as this affects BPC calculations.
Response 4:
Although BPC calculations could be affected by number and location of cores obtained based on biopsy protocol consistency, this issue was not evaluated because locations of obtained cores were not available in all cases; nevertheless, it was not the aim of the study. Accordingly, this limit has been added to the discussion section with the paragraph listing the limits of the study, as well.
Comment 5: Consider the impact of potential multicollinearity between BPC and Id-BPC in multivariate models.
Response 5: Because of potential multicollinearity between BPC and Id-BPC, we did not evaluate both factors simultaneously in multivariate models; likewise, Id-BPC was always a stronger predictor than BPC in both univariate and multivariate models, as shown in Tables 1 and 2, respectively. Accordingly, this point has been included in the 3.2 section and in the discussion, as well.
Comment 6: Tables lack explicit p-values for some comparisons; ensure all relevant statistics are reported.
Response 6: We checked all tables and all relevant statistic relating to comparisons has been assessed and reported.
Comment 7: Discuss the clinical meaning of a 3.5-fold increase in PLNI risk with Id-BPC greater than 1.0 mL, including patient management implications.
Response 7:
So far, our results have practical implications for managing EAU high-risk patients in daily practice for Id-BPC is a simple tool which may be applied when consulting patients; accordingly,
we started from theoretical assumptions relating Id-BPC to the most adverse outcome after surgery; moreover, the study demonstrated that the cutoff of 1.0 %/mL stratified two categories of subjects of whom the one with Id-BPC  1.0 %/mL showing a 3.5-fold increase in PLNI including 23.2% of the EAU high-risk population who need extended counselling with combined aggressive treatments including extended pelvic lymph node dissection for accurate anatomical staging for surgery or androgen blockade associated extended radiation of both the prostate and the pelvis, as well. These features have been integrated into the discussion section, accordingly.
Comment 8: Verify that all cited references are up-to-date and cover the latest guidelines and studies relevant to high-risk prostate cancer.
Response 8: We have verified that all cited references are updated to EAU and NCCN guidelines which are the leading recommendations for dealing with high-risk prostate cancer, as well.
Comment 9: Include a more detailed institutional review board approval section, specifying approval date and reference number.
Response 9: We provided more detailed institutional review board approval section , accordingly.
Reviewer 2 Report
Comments and Suggestions for Authors
This manuscript presents a clinically relevant and statistically robust analysis of the predictive value of index cancer density (Id-BPC) compared to the percentage of biopsy-positive cores (BPC) in patients with EAU high-risk prostate cancer undergoing robot-assisted radical prostatectomy. The study benefits from a relatively large cohort and clear statistical modeling.
The main strength of the study is the demonstration that Id-BPC is a stronger and independent predictor of pelvic lymph node invasion (PLNI) than BPC alone, with potential implications for preoperative risk stratification and future predictive modeling.
The manuscript is clearly written and well-structured. The statistical methods are appropriate, and the results support the conclusions. The inclusion of multivariable models and exploration of clinical impact strengthens the findings.
Only minor suggestions are recommended:
- Please ensure all abbreviations are defined at first use in the main text.
- Consider a brief clarification in the discussion regarding how Id-BPC may be integrated into future nomograms or AI models.
Overall, this is a valuable contribution to the field and is suitable for publication after minor revision.
Author Response
Comment 1: Please ensure all abbreviations are defined at first use in the main text.
Response 1: These suggestions have been adjusted through the manuscript.
Comment 2: Consider a brief clarification in the discussion regarding how Id-BPC may be integrated into future nomograms or AI models.
Response 2: EAU high-risk PCa is a pivotal prognostic class which occults a complex heterogenous biological system needing stratification in order to deliver appropriate sequential combined treatments, as well. Likewise, our study showed that 23% of the high-risk population harbored pelvic lymph node metastases which were predicted by standard clinical parameters including PSA > 20 ng/mL, ISUP > 3 and BPC; likewise, Id-BPC demonstrated stronger prediction than the latter in multivariate model, as shown in Table 2; moreover, it allowed effective patient stratification when categorized at the median level. Interestingly, these findings are critical for understanding the prostate cancer high-risk system; accordingly, high PSA levels related to the metastatic load at pelvic lymph nodes while undifferentiated cancers associated adverse tumor stage including extracapsular extension and seminal vesicle invasion which is closed to related to PLNI according to the complex anatomy associated the lymphatic pathway originating from the prostate gland; likewise, PLNI was also predicted by both continuous and categorized Id-BPC which is a more impactful factor than BPC itself for being a bidimensional variable relating BPC to the volume of the prostate thus defining a more aggressive biology within the EAU high-risk system, as well. As a result, Id-BPC should be included in multivariate models which specifically evaluate the risk of PLNI in the prostate cancer high-risk population; accordingly, it can be included in dedicated nomograms as well as in artificial intelligence (AI) networks thus impacting more then those including only BPC on clinical decisions for the machines being able to test and learn several patterns that may present effective solutions which need to be assessed by the multidisciplinary PCa team in order to plan appropriate and effective decisions when counselling patients presenting with the unfavorable prognostic risk class, as well.
These considerations have been included into the discussion section which amplifies and improves the impactful findings of the study.
Round 2
Reviewer 1 Report
Comments and Suggestions for Authors
Accept in present form